# Mediterranean and Black Sea Monstrilloid Copepods (Copepoda: Monstrilloida): Rediscovering the Diversity of Transient Zooplankters

**Eduardo Suárez-Morales** [1,*] **and Mark J. Grygier** [2]

1    El Colegio de la Frontera Sur (ECOSUR), Chetumal 77014, Mexico
2    Center of Excellence for the Oceans, National Taiwan Ocean University, No. 2, Beining Rd.,
     Zhongzheng Distr., Keelung 202301, Taiwan; thecostracans@gmail.com
*    Correspondence: esuarez@ecosur.mx

**Abstract:** Monstrilloids are copepods that live freely in plankton without feeding but have parasitic immature stages that develop within infected benthic molluscs and polychaetes. Because of their incompletely known life cycles and the difficulty of matching conspecific males and females, it has been difficult to assess their true diversity anywhere on earth. The monstrilloid fauna of the Mediterranean and Black seas (MBS) has been investigated for over 140 years, during which time four phases of study can be recognized. The initial list of MBS monstrilloids recorded during the first phase (1877–1893) grew only slowly for decades afterwards during the second phase (1895–1952) because of patchy sampling and a dearth of formal taxonomic descriptions. The third phase (1957–1986) featured little new work at all. During the most recent fourth phase since 1992, a reappraisal with heed to nomenclatural rules and upgraded descriptive standards has led to the realization that many nominal species of MBS monstrilloids are invalid or doubtful. Furthermore, some that have been frequently recorded, such as *Monstrilla grandis*, *Cymbasoma longispinosum*, and *C. rigidum*, may actually be undescribed representatives of widespread species groups. We provide an updated annotated checklist of MBS monstrilloids that includes 21 supposedly valid nominal species or species-groups. This rather high regional diversity will likely grow if future zooplankton surveys in the highly heterogeneous and extensive coastal systems of the MBS pay due attention to this intriguing group of copepods.

**Keywords:** zooplankton; monstrilloids; copepods; diversity; distribution

## 1. Introduction

Among the 10 currently recognized orders of Copepoda [1], the Calanoida comprise the most diverse and abundant group of marine zooplankton [2]. Calanoids are highly diverse in the Mediterranean Sea, where nearly 440 species have been recorded [3]; when considering the non-calanoid planktonic copepods as well, the number of species occurring there rises to more than 590 [4,5].

The Monstrilloida, which are one of the least studied copepod orders, parasitize benthic molluscs and polychaetes as larvae while the better-known non-feeding and free-swimming adults can be found in plankton samples. Comprising the single family Monstrillidae, they appear to be most abundant and diverse in coastal habitats and over coral reefs [6,7]. The main aspects of their taxonomic diversity have been addressed [7,8]. Currently, the order and family are known to contain over 155 nominal species [4,7–9]) in seven valid genera [10,11], although doubts about which genera are valid continue to be raised [12,13].

As a group, monstrilloids have been observed since the earliest marine planktological surveys carried out during the 19th century; their literature begins around 1845 [14].

Taxonomic exploration is still being done in different regions of the world, including previously unstudied areas like Australia [15,16] and Korea [12,13,17–20].

The Mediterranean-Black Sea (MBS) monstrilloid fauna has been considered among the best known worldwide [7]. The most up-to-date regional list [5] contains 28 nominal species in two genera. It partly reflects recent revisionary work conducted in accordance with current taxonomic and nomenclatural standards [11,21–28], but other records must still be reexamined. Here, we present an historical overview of monstrilloid studies in the MBS, comment on certain aspects of the reported diversity there, and provide a revised and updated regional checklist with annotations.

## 2. Materials and Methods

Based on a survey of the regional literature on Monstrilloida, for which several previous compendia (e.g., [4,5,14,29–31]) served as starting points, we divided the advance of knowledge about monstrilloid diversity and distribution in the MBS into four historical phases. Table 1 shows the additions of nominal species to the MBS monstrilloid inventory during these phases. The list of 21 species for the most recent, fourth phase excludes all invalid or doubtful earlier species records and indicates (as "cf.") those that might include undescribed members of species complexes; it thus serves as a revised and updated checklist of the MBS-inhabiting monstrilloids. In the following account, depending on context, generic assignments of the mentioned species are often updated (e.g., from invalid *Thaumaleus* to valid *Cymbasoma*), and their spellings are corrected, without comment. The generic abbreviation "*M.*" is only used for *Monstrilla*, and "*C.*" only for *Cymbasoma*.

**Table 1.** Original records of nominal species of monstrilloid copepods recorded from the Mediterranean and Black seas in primary and secondary literature during each of three historical phases of study (1877–1893, 1895–1952, 1957–1986), with a checklist of those currently accepted as valid plus additional species recorded since 1992. Spellings have been corrected. Species originally recorded as *Thaumaleus* are here listed as *Cymbasoma*, and all species of *Caromiobenella* were originally recorded as *Monstrilla*.

| Phase 1: 1877–1893 | Phase 2: 1895–1952 | Phase 3: 1957–1986 | Phase 4: 1992–2020 (Current Checklist) |
|---|---|---|---|
| *Cymbasoma claparedii* *C. herdmani* | *Cymbasoma claparedii* | *Cymbasoma longispinosum* | *Cymbasoma clauderazoulsi* Suárez-Morales, Goruppi, de Olazabal and Tirelli, 2017 |
| *C. longispinosum* | *C. longispinosum* | *C. rigidum* | *C.* cf. *longispinosum* (Bourne, 1890) |
| *C. reticulatum* | *C. rigidum* | *C. tenue* | *C. mediterraneum* Suárez-Morales, Goruppi, de Olazabal and Tirelli, 2017 |
| *C. rigidum* | *C. thompsonii* | *C. thompsonii* | *C. nicolettae* Suárez-Morales, 2002 *C. pseudobidentatum* Suárez-Morales, Goruppi, de Olazabal and Tirelli |
| *Monstrilla gracilicauda* | *Monstrilla gracilicauda* | *Monstrilla grandis* | *C.* cf. *rigidum* Thompson, 1888 |
| *M. intermedia* (purported synonym of *M. grandis*; suppressed by ICZN) | *M. grandis* (original source of record unclear) | *M. leucopis* (likely erroneous) | *C. reticulatum* (Giesbrecht, 1893) |
| *M. longissima* (*nomen nudum*) | *M. helgolandica* | *M. longicornis* | *C. sinopense* Üstün, Terbiyik and Suárez-Morales, 1914 |
| *M. longiremis* | *M. longicornis* (original source of record unclear) | *M. longiremis* | *C. specchii* Suárez-Morales, Goruppi, de Olazabal and Tirelli, 2017 |
| *M. pontica* (insufficiently known) | *M. longiremis* | *M. tumorifrons* (*nomen nudum*; later made available) | *C. tenue* (Isaac, 1975) *C. tergestinum* Suárez-Morales, Goruppi, de Olazabal and Tirelli |
| | *M. ostroumowi* (now in *Cymbasoma*; possibly a synonym of *C. rigidum*) | *Monstrillopsis angustipes* (*nomen nudum*) | *C. tumorifrons* Suárez-Morales, 1999 *C. turcorum* Suárez-Morales and Üstün, 2018 |

**Table 1.** *Cont.*

| Phase 1: 1877–1893 | Phase 2: 1895–1952 | Phase 3: 1957–1986 | Phase 4: 1992–2020 (Current Checklist) |
|---|---|---|---|
| | *M. serricornis* (supposed junior synonym of *M. helgolandica*; original source of record unclear) | | *Monstrilla ghirardellii* Suárez-Morales, Goruppi, de Olazabal and Tirelli, 2017 |
| | *Monstrillopsis dubia* (species complex; original source of record unclear) | | *M. grandis* Giesbrecht, 1891 |
| | *Monstrillopsis zernowi* (probably a *Monstrilla*) | | *M. longicornis* Thompson, 1890 *M. longiremis* Giesbrecht, 1893 |
| | | | *Monstrillopsis pontoeuxinensis* Suárez-Morales and Üstün, 2018 |
| | | | *Monstrillopsis zernowi* Dolgopolskaya, 1948 |
| | | | *Caromiobenella* cf. *helgolandica* (Claus, 1863) |
| | | | *Ca. pygmaea* (Suárez-Morales, 2000) |

## 3. Results and Discussion

### 3.1. Species Diversity of the Monstrilloida

Currently, there are about 155 nominal species in this copepod order, but the number of valid species is somewhat smaller. The taxonomic and nomenclatural problems exposed in different works [7,8,14,32,33] have revealed many unlikely species records around the world, and especially a number of improbable cosmopolitan distributions. In earlier summaries of the group's diversity [7,8], 116–125 species were considered valid, with the following numbers of species in each then-recognized genus: *Monstrilla* (56 species), *Cymbasoma* (41), *Monstrillopsis* (12), and *Maemonstrilla* (7). Since then, the discovery of three additional genera of monstrilloids [11,12,15] and many undescribed species, as well as the delineation of diverse species complexes, has resulted from recent exploration of unstudied areas as noted above. This has led to consistent growth in regional and world lists of Monstrilloida. The same can be expected in the MBS. Until very recently, only species of the genera *Cymbasoma* and *Monstrilla* had been confirmed as inhabiting MBS waters. *Monstrillopsis zernowi*, potentially a representative of a third genus, was described from the Black Sea [34], but it is probably a species of *Monstrilla* [35]. We do not know of any basis in the primary literature for the occasional mentions of *Monstrillopsis dubia* as a Mediterranean species in secondary sources [29,36]. The description of *Monstrillopsis pontoeuxinensis* from the Black Sea [35] has, however, confirmed the presence of this genus in the MBS region. In addition, the recently proposed [12] new and widespread genus *Caromiobenella* includes at least two MBS species, the former *Monstrilla helgolandica* and *M. pygmaea*, thereby adding a fourth genus to the currently known monstrilloid fauna of these seas (Table 1).

### 3.2. Mediterranean-Black Sea Diversity of the Monstrilloida

MBS waters represent a highly suitable environment for monstrilloids. They comprise a vast set of marine coastal ecosystems, with an irregular, deeply indented coastline, resulting in an extremely rich variety of coastal habitats. The Mediterranean alone is said to host 4% to 18% of the world's marine biodiversity [37,38]. In considering the world fauna of Monstrilloida, it has been suggested [7] that the regions with the highest known species richness were the European waters of the North Atlantic (32 species), the Caribbean Sea and Gulf of Mexico (24), the Mediterranean-Black Sea region (19), the Indonesia-Malaysia-Philippines region (17), the waters around Japan (17), and the Brazil-Argentine area (16). An update of these data is needed because of the increasing number of species described

in some of these regions. With this in mind, here, we review and update what is known concerning MBS monstrilloid diversity.

### 3.3. MBS Monstrilloid Diversity, Historical Account

The monstrilloid fauna of the MBS region has been investigated for almost 145 years. Four phases in the region's history of monstrilloid studies can be distinguished. The first preliminary phase (1877–1893) began with Kriczagin's (1877) [39] description of three supposedly new species: *Monstrilla intermedia* and *M. pontica* from five sites along the northeastern shore of the Black Sea and *M. longissima* from an unspecified Mediterranean site. *Monstrilla intermedia* was suppressed for purposes of priority in favor of *M. grandis* (for which see below) by the International Commission on Zoological Nomenclature [40]. Isaac (1975) [41] tentatively synonymized *M. longissima* with *Cymbasoma longispinosum*, but Grygier (1995) [14] considered the former "evidently a *nomen nudum*". *Monstrilla pontica*, currently *Cymbasoma ponticum*, has never been restudied and is excluded from our final checklist (Table 1) along with Kriczagin's other two species because its identity remains uncertain.

Subsequently, *Cymbasoma herdmani* was described in part from the Mediterranean Sea at Malta [42], where *C. rigidum* was also recorded [43]. The type series of *C. herdmani* included a specimen from North Wales as well as one from Malta; in context it seems likely that the former specimen served as the basis for the former paper's illustrations, but no lectotype has ever been selected. In British waters, *C. herdmani* was soon relegated to the synonymy of *Monstrilla anglica* [44–46], but the identity of the syntype from Malta remains unresolved [46]. Because *M. anglica* has never been recorded anew from the Mediterranean under its own name, there is a chance that the Maltese *C. herdmani* is not conspecific with it. Wilhelm Giesbrecht's (1893) [46] seminal work on the copepods of the Gulf of Naples, which has served as a principal reference work for all later researchers on MBS monstrilloids, included records from Naples of three species of *Thaumaleus* (*T. longispinosus*, *T. claparedii*, and *T. reticulatus*, all now in *Cymbasoma*) and two of *Monstrilla* (*M. gracilicauda* and *M. longiremis*). Giesbrecht thus brought the total number of nominal species recorded from the MBS region during this early period to 11, including those later judged invalid or doubtful (Table 1).

During the second phase (1895–1952), exploratory planktological work continued at various places in the MBS, but few taxonomic descriptions ensued. In the immediate post-Giesbrecht period, Karavayev (1895) [47] described *M. ostroumowi*—possibly a synonym of *C. rigidum* [41,48] from Sevastopol Bay in the northeastern Black Sea; Graeffe (1900) [49] reported *Cymbasoma rigidum* from the Gulf of Trieste; *M. longiremis* and *C. longispinosum* were recorded from the Dalmatian coast of the Adriatic Sea at Tiesno (currently Tisno) and at Vodice and Rieka (currently Rijeka), respectively [50]; and *C. thompsonii* was reported from the Lagoon of Venice [51,52]. These early Adriatic records were later compiled [53]. In the meantime, Lo Bianco (1903) [54] reported *C. longispinosum* and unidentified monstrillids referred to as "*Thaumaleus* sp." from three sites in the vicinity of Capri, Italy, and *Monstrilla grandis* was reported from the Black Sea coast of Bulgaria [55]. Later, there were various reports [56–60] of *C. longispinosum* from "the Mediterranean", *C. rigidum* from Monaco, and *M. helgolandica*? (later with no question mark), *Monstrilla longiremis*, *M. gracilicauda*, *M.* sp., *C. longispinosum*, *C. claparedii*, and *C. rigidum* from the Bay of Algiers in North Africa [29,30]. *Monstrilla longicornis* and *Monstrillopsis dubia* were mentioned as Mediterranean species [30] while *M. serricornis* and *M. grandis* were included in a list of North African species, but we do not know what primary literature these records were based on, or what purported synonymies they may reflect. It is clear, however, that the latter authors treated *M. serricornis* as a synonym of *M. helgolandica*, thus as an invalid name.

According to a summary of the Adriatic plankton fauna [61], *Monstrilla longiremis* was recorded from the Adriatic at Trogir, Yugoslavia [62], and also from Kastela Bay [63], and *C. longispinosum* is mentioned as abundant in the northern Adriatic. Studies were then interrupted by war, after which *Monstrillopsis zernowi* was described [34] from Yarylgach

Bay in western Crimea, northern Black Sea; this species is most likely actually a species of *Monstrilla* [35]. Dolgopolskaya (1948) [34] also recorded *M. grandis*, *M. helgolandica*, and *C. longispinosum* from there and/or Sevastopol Bay. Subsequently, *C. longispinosum* was reported from the Étang de Thau, a large lagoon on the French Mediterranean coast [64], and this long second historical phase closed with a list of North African copepods, including five nominal species of monstrilloids [30]. By our count, 12 nominal species, including eight not previously recorded and several judged by us as problematic, were recorded from the MBS during the second phase of study (Table 1).

The following third historical phase of monstrilloid studies in the MBS (1957–1986) was characterized by infrequent observations and little formal taxonomic treatment. Only four additional names are found among the ten species of monstrilloids recorded from the MBS in primary sources during this period (see Table 1). This phase opened with a mention of the occurrence of *M. grandis* in Varna Bay, Bulgaria [65], and with a widely known manual of Mediterranean planktology [31], in which just six monstrilloid species were illustrated (including again *M. longicornis*, the basis for which is unclear). *Cymbasoma rigidum* and *C. thompsonii* were reported from Constanta on Romania's Black Sea coast [48] and *M. grandis* was recorded from Marseilles [66]; *M. leucopis* was recorded from Villefranche [67], this last probably being a misidentification because *M. leucopis* appears to be restricted to Norway [26]. *Monstrilla longiremis* was reported from Yugoslavia's Bay of Mali Ston [68], and unidentified monstrilloids were reported from Livorno, Rapallo, and San Remo in northern Italy [69]. In the Levantine Basin, *C. longispinosum*, *C. rigidum*, and *Monstrilla* sp. were also reported [70]. In addition, *M. longicornis* and *C. longispinosum* were recorded from Marseilles [71].

Isaac's (1975) [40] introduction of the names *Monstrilla tumorifrons* and *Monstrillopsis angustipes* as self-declared *nomina nuda* in a key deserves special mention. Relatively full descriptions of both species were provided in an unpublished dissertation [72], and both came from Emborios Bay, Aegean Sea (i.e., the island of Chios, Greece). According to a recent nomenclatural review [73], the former name was later made available [22], but the latter remains unavailable. Pending a restudy of Isaac's material and validation of the name, *Monstrillopsis angustipes* is omitted from the final checklist herein (Table 1).

Razouls and Durand (1991) [36] closed this third period of study with an inventory of the then known Mediterranean copepod fauna. Of the 11 nominal species of monstrilloids they listed, three had not appeared to our knowledge in previous primary literature: *Haemocera danae*; *Monstrilla conjunctiva*, which is a supposed senior synonym of *M. leucopis* [74]; and *Monstrillopsis dubia*, a name perhaps carried forward from a previous work [29].

The most recent, fourth historical phase of monstrilloid research in the MBS (1992–2020) has featured a revival of detailed taxonomic work. Simple distribution records in different MBS locations include (presumably different) *Monstrilla* sp. at Tunis in North Africa [75], in a submarine cave of Italy's Salento Peninsula [76] and off the Kerch Peninsula of eastern Crimea [77], *Monstrillopsis zernowi* from a coastal bay near Sevastopol, Crimea [78], *Monstrilla grandis* from Toulon Bay, France [23], and *Monstrilla* sp. and unidentified monstrilloids respectively in plankton and in gut contents of pipefish at a total of four nearshore sites in Izmir Bay on the Aegean coast of Turkey [79,80].

Revisionary work has so far focused on two older species. Redescriptions of both sexes of *Cymbasoma tenue* [21,23] were based on specimens from Toulon Bay, France; the latter paper also included a description of *Monstrilla pygmaea* (currently in *Caromiobenella*) from the same bay. *Cymbasoma tumorifrons* was also redescribed [22,24] based on original material from the island of Chios, Greece, and a newly caught female from Toulon Bay, while *C. nicolettae* was described from Toulon Bay [24]. Later, while studying monstrilloids from the Gulf of Trieste in the northern Adriatic Sea, females of *C. tumorifrons* from Greece and Toulon were reassigned to the new species *C. mediterraneum*, described from Trieste [28]. This was one of six new species from the Gulf of Trieste, five of *Cymbasoma* plus *M. ghirardellii*, that were described in the same work (see Table 1; another species remains undescribed), and the paper also reported very high local abundances of *M. grandis*. Other

recent additions to the MBS monstrilloid fauna are the three new species described from the southern coast of the Black Sea at Sinop, Turkey: *C. sinopense*, *C. turcorum*, and *Monstrillopsis pontoeuxinensis* [35,81,82].

A map of all the MBS collection sites mentioned in the preceding historical review (Figure 1) shows that the sampling effort for monstrilloid copepods has been limited in the Mediterranean Sea largely to the French coast, the Bay of Algiers, the Gulf of Naples, and the Dalmatian coast and Trieste in the Adriatic Sea, in addition to scattered sites (notably around Crimea and at Sinop) in the Black Sea. Many coastal areas in both seas have produced only unidentified monstrilloids or remain completely unsurveyed for this group. The mainland Greek, Albanian, Spanish, Moroccan, Libyan, Egyptian, Israeli, Lebanese, southern Turkish, and Georgian coastlines, as well as all major Mediterranean islands, are blank. A recent comprehensive review of the Iberian non-calanoid copepods [83], including both the Atlantic and Mediterranean fauna, listed seven species each of *Cymbasoma* and *Monstrilla* based entirely on previous reviews. Among them, only *C. nicolettae*, which has been recorded from Toulon Bay in France, but not Spain [24], was explicitly presented as a Mediterranean form.

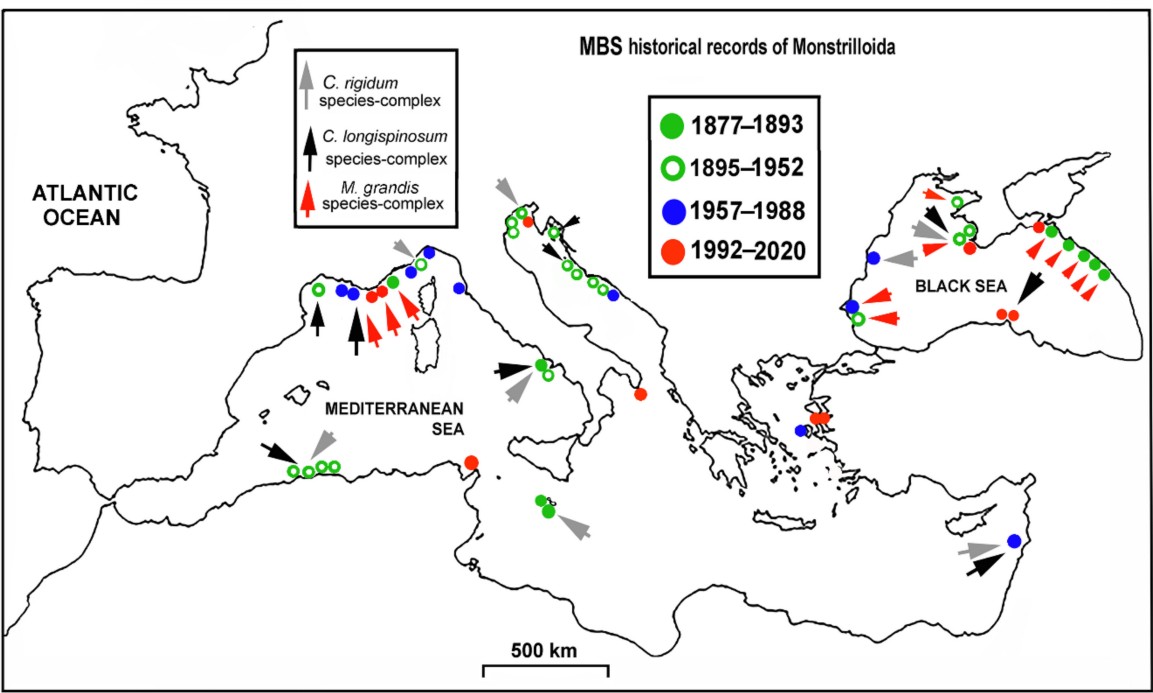

**Figure 1.** Historical and geographic distribution of MBS records of monstrilloid copepods between 1877 and 2020. Color dots refer to each of four historical periods. Black, gray, and red arrows represent the locations of species assignable to one of three species-complexes known to occur in the MBS region.

Much of the early literature on monstrilloids, at least up until Wilhelm Giesbrecht's (1893) [46] work, lacked detailed descriptions and illustrations of the species that were treated and named therein. In many cases, this has made subsequent identification difficult and untrustworthy, especially when a given nominal species has later been recorded from widely different geographic areas. Some such putative "species" have by now been recognized as species complexes [7,8,17,84]. The once supposedly cosmopolitan *Monstrillopsis dubia* is now deemed a complex of several different species [85]. A similar situation has been noted for *Caromiobenella* (formerly *Monstrilla*) *helgolandica* [12], *Monstrilla grandis* [17,82], and two species of *Cymbasoma* (*C. longispinosum* and *C. rigidum*; see below), all of which seemingly comprise many undescribed taxa.

### 3.4. Analysis of Selected Species-Groups

#### 3.4.1. *Cymbasoma longispinosum* Species-Group

After Bourne's (1890) [44] original description of *C. longispinosum* from the English Channel, this nominal species was recorded from many different parts of northern and western Europe, the MBS region, West Africa, the Red Sea and Arabian Gulf, India, Australia, the Philippines, Japan and Brazil [14,86]. Aside from the dubious proposition that a single monstrilloid species could be so widespread, some authors noted morphological differences between their specimens and the original description. It was later realized that this nominal species consists of a group of morphologically close species, each with a limited geographic distribution and subtle but consistent diagnostic distinctions [81,86,87]. Together with the previously described *C. morii* from Japan [33], they constitute a large species-group, eight additional members of which have recently been described [16,86,87]. As was recounted above and also shown in Figure 1, there have been many records of supposed *C. longispinosum* in different parts of the MBS region, but owing to a lack of descriptive information, their true identity and diversity remain unconfirmed. One new member of this species-group, (*C. sinopense*), was recently described from the Turkish Black Sea coast [81]; the same paper noted that this Black Sea population differs in various respects from supposed *C. longispinosum* in the Mediterranean, at least as described by W. Giesbrecht (1893) [46].

#### 3.4.2. *Cymbasoma rigidum* Species-Group

This species was described by Thompson (1888) from the Canary Islands [42]. At least three distinct morphotypes have been distinguished among the many illustrated records of *C. rigidum* from different parts of the world [25]. This variability, the uncertain identity and unclear morphology of the original specimen, and the wide geographic scope of records of *C. rigidum* (for the MBS region, see Figure 1) sow doubt about whether all records pertain to one species, and if not, how widely the holotype's species is distributed. We suspect that this nominal species constitutes a species-complex with several undescribed taxa. Suárez-Morales (2006) [25] noted several differences between MBS *C. rigidum* and other European and Asian sites. Graeffe's (1900) [49] early record of *C. rigidum* from the Gulf of Trieste might actually pertain to either *C. tergestinum* or *C. specchi*, which were later described from that region [28]. In the Black Sea, *M. ostroumowi* has been recognized as a synonym of *C. rigidum* (e.g., [48]) but may possibly be a Black Sea or MBS endemic.

#### 3.4.3. *Monstrilla grandis* Species-Group

This presumably widespread species was originally described by Giesbrecht (1891) from the southwestern Atlantic at 49° S, 65° W, with no more than a brief diagnosis of the female [88]; both sexes were later described in detail based on this Atlantic material [46]. Suárez-Morales (2000) [23] provided a complementary description of the male based on specimens from Toulon Bay in the Mediterranean. In the meantime, as was noted above (see also Figure 1), purportedly the same species had been recorded from different areas of, first, the Black Sea (originally under the name *M. intermedia*), and later, the Mediterranean. As summarized by Grygier (1995) [14] and newly recorded by more recent authors (e.g., [12,84,89,90]), *M. grandis* has also been reported from other parts of the world: the northeastern Atlantic from the North Sea to Morocco, East Asian waters of Russia, China, Japan, and Korea; Barbados, Puerto Rico, and Costa Rica in the Caribbean region, and Brazil and Chile. The geographically closest records to the type locality are from coastal and offshore areas of Argentina [84,89,91]. Morphological differences have been detected among both males and females from different areas [17,84], and morphometric differences exist in both sexes from the Adriatic Sea with respect to specimens recorded from other regions [28]. Such comparative data support the contention that undescribed species may be included in the nominal species *M. grandis* [28].

## 4. Conclusions

We count 20 species of monstrilloid copepods as confirmed inhabitants of the Mediterranean and Black seas (Table 1). Our chronologically categorized distribution map of records of Monstrilloida from this región (Figure 1) emphasizes in a graphic way just how small and scattered the well-sampled shorelines are, compared to the entire array of complex marine coastal systems present in these seas. New regional sampling efforts will surely result in a continued and highly significant lengthening of the lists of planktonic copepods, including monstrilloids, that occur there.

**Author Contributions:** Both authors contributed equally to the study's concept and design, the literature review, and the writing and editing of the text. All authors have read and agreed to the published version of the manuscript.

**Funding:** This research received no external funding. General support was provided by El Colegio de la Frontera Sur (ECOSUR) to ES-M, and MJG's work was enabled by support to National Taiwan Ocean University's Center of Excellence for the Oceans by the Featured Areas Research Center Program within the Taiwan Ministry of Education's Higher Education Sprout Project.

**Institutional Review Board Statement:** Not applicable.

**Informed Consent Statement:** Not applicable.

**Data Availability Statement:** The data presented in this study are available on request from the corresponding author.

**Acknowledgments:** E.S.-M. thanks the El Colegio de la Frontera Sur (ECOSUR) for allowing him to work from home during the past year.

**Conflicts of Interest:** The authors declare that they have no conflict of interest.

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
