# Peer review of "Mediterranean and Black Sea Monstrilloid Copepods (Copepoda: Monstrilloida): Rediscovering the Diversity of Transient Zooplankters"

_water, doi:10.3390/w13081036_

Round 1
Reviewer 1 Report
Review of the manuscript Mediterranean monstrilloid copepods (Copepoda: Monstrilloida): rediscovering the diversity of transient zooplankters by Eduardo Suárez-Morales and Mark J. Grygier
The purpose of this work is to review the current scattered knowledge of the diversity of Mediterranean monstrilloids. The authors described chronologically the progress of knowledge on certain species in the region, indicating nomenclature changes of individual species and uncertain species. This is a very valuable and interesting study/review, and considering that there are few specialists in this group of copepods in the world, this work indicates the gaps in knowledge and is the basis for further work on the monstrilloid in the Black Sea and the Mediterranean. I recommend publishing the paper in the present form. I just want to put here a small remark, that maybe it would be better for the readability of the work if the authors will edit the first sentence of the conclusion (line 321-324). And probably the word "región" (line 325) needs correction.
Reviewer 2 Report
Excellent paper, in my opinion it can be published without any changes.
Reviewer 3 Report
Review
Paper title: Mediterranean Monstrilloid Copepods (Copepoda: Monstrilloida): Rediscovering the Diversity of Transient Zooplankters.
The Mediterranean Sea is an important area in terms of fish production, aquaculture activities, potential for tourism, and ecosystem services. However, little is known about the current diversity of Monstrilloid copepods which are difficult to investigate. The authors are well-known specialists in this field. In the present paper, they analyzed historical data and provided a checklist of the monstrilloid fauna of the Mediterranean and Black seas. This information updates inventories and databases of copepods from the study area, provide a baseline for further biodiversity studies, and may be important for monitoring pelagic ecosystems in the Mediterranean area.
All these reasons explain the relevance of the paper by Suárez-Morales E. and Grygier M.J. submitted to "Water".
General scores.
The data presented by the authors are significant. The authors considered and comprehended relevant literature sources to provide an excellent overview in this field. We authors conducted careful work which will attract the attention of a wide range of specialists including taxonomists and planktonologists.
I recommend this paper for publication after minor revisions.
Specific comments.
Title. The authors considered the Mediterranean-Black Sea fauna whereas only "Mediterranean" occurs in the title. I suggest to change the title according to the data presented in the text.
L 8-9. Please, insert your correspondence information.
Citations must be formatted according to Instructions for authors.
L 73. I suggest to change M. to "M." and C. to "C."
L 75. This table is the same as presented in the Supplementary material. Thus, Supplementary material is not necessary.
L 81. Change " Results" to "Results and Discussion"
L 116-118 These two sentences " An update of these data…" is a repetition of previous statements. I suggest to delete these.
L 119. Change " MBS" to " Mediterranean-Black Sea"
L 199. " nomina nuda" should be italicized.
L 267. " Cymbasoma Longispinosum" should be italicized.
L 286. " Cymbasoma Rigidum" should be italicized.
L 300. " Monstrilla Grandis" should be italicized.
L 308-312. The authors should include in this list a recent record by Bonk et al. (2019) who found M. grandis in waters off the Kamchatka Peninsula, Sea of Okhotsk, Russia.
Reference:
Bonk T.V., Sushkevich N.S., Lozovoy A.P. The first foundation the Monstrilloida (Copepoda) species in Okhotsk Sea near Kamchatka shore. In: Conservation of biodiversity of Kamchatka and coastal waters: materials of the XX International scientific conference, dedicated to the 150th anniversary of academic V.L. Komarov’s birthday; Tokranov A.M. (Ed). Kamchatpress: Petropavlovsk-Kamchatsky, Russia, 2019. pp. 157–159 (in Russian).
https://www.elibrary.ru/item.asp?id=41756138
See attached file.
L 321-323. " (This number assumes that the two missing species of 2017 Cymbasoma are added to Table 1, and that M. longicornis remains in the final checklist (although it is really based on just one unconfirmed record).)". I suggest to delete this insertion.
L 337-349. Delete these incomplete sections.
